# External load transition practices from pre-season to in-season. A case study in elite female professional soccer players

Rafael Oliveira[1,2,3]*, Ryland Morgans[4], Renato Fernandes[1,2], João Paulo Brito[1,2,3], Mário C. Espada[5,6,7,8,9], Fernando J. Santos[5,6,10]

1 Santarém Polytechnic University, School of Sport, Rio Maior, Portugal, 2 Life Quality Research Centre (CIEQV), Santarém Polytechnic University, Rio Maior, Portugal, 3 Research Center in Sport Sciences, Health Sciences and Human Development (CIDESD), Santarém Polytechnic University, Rio Maior, Portugal, 4 School of Sport and Health Sciences, Cardiff Metropolitan University, Cardiff, United Kingdom, 5 Life Quality Research Centre (CIEQV-IP Setúbal), Campus do IPS–Estefanilha, Setúbal, Portugal, 6 Instituto Politécnico de Setúbal, Escola Superior de Educação, Setúbal, Portugal, 7 Faculdade de Motricidade Humana, Interdisciplinary Centre for the Study of Human Performance (CIPER), Universidade de Lisboa, Lisbon, Portugal, 8 SPRINT Sport Physical Activity and Health Research & Innovation Center / Centro de Investigação e Inovação em Desporto Atividade Física e Saúde, Rio Maior, Portugal, 9 Comprehensive Health Research Centre (CHRC), Universidade de Évora, Évora, Portugal, 10 Faculdade de Motricidade Humana, Universidade de Lisboa, Lisbon, Portugal

* rafaeloliveira@esdrm.ipsantarem.pt

## Abstract

The study aim was to compare the external load during varying microcycles (M1-M4 during pre-season and M5 during the in-season) in elite female Portuguese soccer players and to describe external load variations between differing Ms. Fourteen first-team players participated in the study (age 23.29 ± 3.19 years, weight 59.14 ± 6.87 kg, height 1.66 ± 0.08 m). Load measures included total distance, high-speed running (HSR) distance ($\geq$15 km/h), number of accelerations and decelerations [acceleration 1 (ACC1), >1–2 m/s; acceleration 2 (ACC2), >2–3 m/s; acceleration 3 (ACC3), >3–4 m/s; acceleration 4 (ACC4), >4 m/s] and decelerations [deceleration 1 (DEC1), <1–2 m/s; deceleration 2 (DEC2), <2–3 m/s; deceleration 3 (DEC3), <3–4 m/s; deceleration 4 (DEC4), <4 m/s]. M1 showed higher values of total distance, ACC1, ACC2, ACC3, DEC2, DEC3 and DEC4 than M2 (p = <0.001–0.04), although HSR was higher in M2 (p < 0.001). M2 showed lower values of HSR, ACC1, DEC1, DEC2, DEC3 than M3 (p = 0.001–0.04). M3 reported higher values of total distance, ACC1, ACC2, DEC1, DEC2 than M4 (p = <0.001–0.03), while M4 only found higher values of ACC4 and DEC4 than M5 (both, p = 0.01). The highest values occurred in M3 for the majority of external load metrics (except ACC3, ACC4, and DEC4 which were higher in M4). However, during the transition from M4 to M5, only ACC4 and DEC4 decreased with all other measures maintained, thus supporting the notion to maintain similar loading patterns during official competition.

**Data Availability Statement:** Data can be found in the Supporting Information files. It can also be available from the Research Ethics Committee of the Polytechnic Institute of Santarém, Santarém,

Portugal (comissaodeetica@ipsantarem.pt) and by the following link: https://doi.org/10.6084/m9.figshare.27868182.v1.

**Funding:** This work was funded by National Funds by FCT - Foundation for Science and Technology under the following project UIDP/04748/2020 and UIDB/04045/2020 (https://doi.org/10.54499/UIDB/04045/2020). The funders had no role in the design of the study; in the collection, analyses, or interpretation of data; in the writing of the manuscript, or in the decision to publish the results.

**Competing interests:** The authors have declared that no competing interests exist.

# Introduction

Load monitorization allows coaches and staff to improve the individualization of training approaches and stimuli, which consequently supports recovery strategies, thus reducing fatigue and injury risks [1,2]. The practice of load monitorization can be measured through two dimensions: internal [e.g., rating of perceived exertion (RPE), various heart rate derivatives] as a psychophysiological response; and external (e.g., running and accelerometry-based measures), considering the physical work performed [3]. Recently, a systematic review on internal and external training load during varying microcycles (M) and match load practices in female soccer players provided several range values for varying internal and external load (training and match) measures was summarized. Although, no variations between Ms were described [4]. Another systematic review analyzed macrocycles, mesocycles, and Ms training load practices in professional male soccer and found that linear programming was applied, however no study has examined female soccer [5].

Despite previous literature, designing and monitoring load can be dependent on several contextual factors such as the number of matches during the season and current period of the annual plan [6,7]. The annual plan includes the pre-season period which is predominately aimed at returning players to appropriate performance levels of strength and football-fitness [8]. Additionally, the annual plan also includes the in-season which usually attempts to maintain the team physical levels achieved during the pre-season, whilst also improving individuals accordingly [8]. More specifically, Ms in female soccer can include two to 10 training sessions/week during the competitive season [9]. While the pre-season period will involve friendly matches, the in-season will include official matches which can significantly increased loads [10]. Therefore, knowledge detailing pre-season and in-season load strategies can help coaches to appropriately design periodized soccer-specific training routines. Such knowledge can also be beneficial when considering the transition from pre-season to the in-season. Nonetheless, research that compares the load of pre-season and in-season periods is not common in soccer players [8,10,11] and to the authors knowledge, there is no research examining female players.

Although, a better understanding of the periodization practices in professional soccer would be practically helpful for performance and technical coaches to manipulate appropriate training load and volume [12]. Despite this, periodization practices in soccer are frequently based on the coach's philosophy and experience [13]. Therefore, the primary aim of this study was to compare M external load changes (four Ms during pre-season and one in-season M) in elite female Portuguese soccer players. In addition, the secondary aim of the study was to describe and quantify the external load variations between Ms.

# Materials and methods

## Design

This research was an observational study design to examine a single female professional team. Players external load was monitored daily during the five examined Ms (M1—M5) which included four pre-season Ms (M1 –M4) and one M from the in-season period (M5) (26-07-2022 to 04-09-2022). The entire period of analysis included 28 training sessions, three friendly matches, and two official matches from the 2022–23 season. A non-probabilistic sampling protocol was employed to recruit the participants (convenience sample). All external load monitoring quantification was implemented without any interference from the research team. Notably, this study is the second part of previous research [9].

## Participants

Fourteen professional outfield female players (age 23.2±5.9 years, weight 80.3±7.0 kg, height 1.81±0.07 m) from the same Portuguese professional club were included in the present study in which six were defenders, three were midfielders, five were attackers and from the 14, four were national team players [9].

The eligibility criteria was adapted from previous research [9] and included: (i) completed at least 80% of training sessions across the five Ms; (ii) did not use dietary supplements during the study period, (iii) was un-injured during the study; and (iv) did not participate in another training program during the study period. Moreover, the exclusion criteria for the study were: (i) a long-term (three months) injury; (ii) did not present full, complete training data; and (iii) were goalkeepers, due to the large variation in physical demands compared with outfield players.

All data collected resulted from normal analytical procedures regarding player monitoring over the competitive season, nevertheless, prior to data collection, participants were fully informed of the study design and written informed consent was obtained from all participants. The study followed the ethical guidelines for human study as suggested by the Declaration of Helsinki and was approved by the Research Ethics Committee of the Polytechnic Institute of Santarém, Santarém, Portugal (No24-2022ESDRM, 25 July 2022). To ensure confidentiality, all data were anonymized prior to analysis.

## Load quantification

Only team pitch-based training and match sessions were included for analysis. All other sessions, individual training sessions, recovery sessions, and rehabilitation training sessions were excluded [14].

The planning of all soccer content was cyclical in nature and reflective of modern methods of periodization in elite soccer, and thus, the external physical load experienced by players was undulating across a M leading to match-play [15]. The number of days between matches differed [16]. Specifically, there were five Ms (representing six weeks), where the first M (M1) had 10 training sessions and one friendly match, the second M (M2) had five training sessions and one friendly match, the third M (M3) had two training sessions and one friendly match, the fourth M (M4) had six training sessions and one official match, and the fifth M (M5) had five training sessions and one official match. The pre-season period encompassed M1, M2, M3, and M4 while M5 was the initial M of the competitive season.

All training sessions included technical, tactical, physical, and psychological components. All players completed one to two strength- and power gym-based sessions per M incorporating upper and lower body and core exercises [15], although these sessions were not included in the analyses [14].

## External load quantification

A portable 10 Hz GPS device (PlayerTek, Catapult Innovations, Melbourne, Australia) was utilized to produce training and match-play locomotor data. This device also incorporated a tri-axial 100 Hz accelerometer that provided acceleration and deceleration data. These types of GPS devices have recently gained much attention and have been found valid and reliable in team sports when quantifying data [17].

Ten minutes prior to each training session and match, PlayerTek devices were turned on. The devices were placed in a specifically customized vest pocket located on the posterior side of the upper torso fitted tightly to the body, as is typical during training and match-play. The

devices were placed and checked by the same staff member on every occasion, and to avoid inter-unit error, each player wore the same device during the study period [18].

The following metrics were collected for analysis: total distance, HSR ($\geq$15 km/h) [19], numbers of accelerations [acceleration 1 (ACC1), >1–2 m/s; acceleration 2 (ACC2), >2–3 m/s; acceleration 3 (ACC3), >3–4 m/s; acceleration 4 (ACC4), >4 m/s] and decelerations [deceleration 1 (DEC1), <1–2 m/s; deceleration 2 (DEC2), <2–3 m/s; deceleration 3 (DEC3), <3–4 m/s; deceleration 4 (DEC4), <4 m/s] [20]. Relative distances and quantities per minute (m/min and nr/min) for all variables were also examined.

## Statistical analysis

Descriptive statistics are presented as mean ± standard deviation (SD) and percentage change. Normality and homogeneity of the different variables were tested using the Shapiro–Wilk and Levene tests, respectively. Only the total distance of M1 and DEC3 of M2 did not present a normal distribution ($p < 0.05$). Still, considering the centrality trend of the remaining variables, dependent $t$-tests were used to compare: M1 versus M2; M2 vs M3; M3 vs M4; M4 vs M5. When significant results were detected, Hedges effect size (ES) was performed to determine the effect magnitude through the difference of two means divided by the standard deviation from the data, and the following criteria were used: <0.2 = trivial, 0.2 to 0.6 = small effect, 0.6 to 1.2 = moderate effect, 1.2 to 2.0 = large effect, and >2.0 = very large effect [21]. All statistical procedures were executed in the IBM SPSS Statistics for Windows (version 27.0, IBM Corp, Armonk, NY, USA). Significant results were considered at $p < 0.05$.

## Results

Fig 1A–1C represents an overview of descriptive statistics across the different Ms for all GPS measures.

Tables 1–4 present detailed comparisons between M1 and M2 (Table 1), M2 and M3 (Table 2), M3 and M4 (Table 3) and M4 and M5 (Table 4) for the different GPS measures.

## Discussion

The present research aimed to analyze the external load of five Ms (four pre-season Ms and one in-season M) of a professional female soccer team. Considering the descriptive data, some differences in the examined variables between the in-season and pre-season Ms were observed, mainly due to the emphasis of this seasonal period on physical conditioning of players [14]. In the first three pre-season Ms, players covered more distance compared to M4 and M5. Previous research have indicated that total distance is higher during the pre-season M [10,14,22], since the objectives at this season stage are to regain the players' physical levels [23], with physical objectives being prioritized during the initial phases of pre-season (M1-M3) [8].

The present study findings examining load periodization, suggest that load across all metrics was reduced in the last two weeks (M3 and M4) of pre-season as a potential tapering approach [24,25] to commence the initial competitive M. In partial support, other authors found that the number of accelerations/decelerations in training sessions decreased throughout the season [22], albeit the present study had a significantly different physical emphasis during the study period based on the stage of the season. However, in our study, this was only evident for accelerations and decelerations <3m/s. Regarding high-intensity accelerations/decelerations (>3m/s), despite some differences between Ms, lower accelerations/decelerations in zones <3m/s were found. This finding can partly be explained by the coaches of the study team utilized various small-sided games, manipulating short spaces and/or with the use of goalkeepers [26,27], aimed at promoting acceleration/decelerations, braking, jumping,

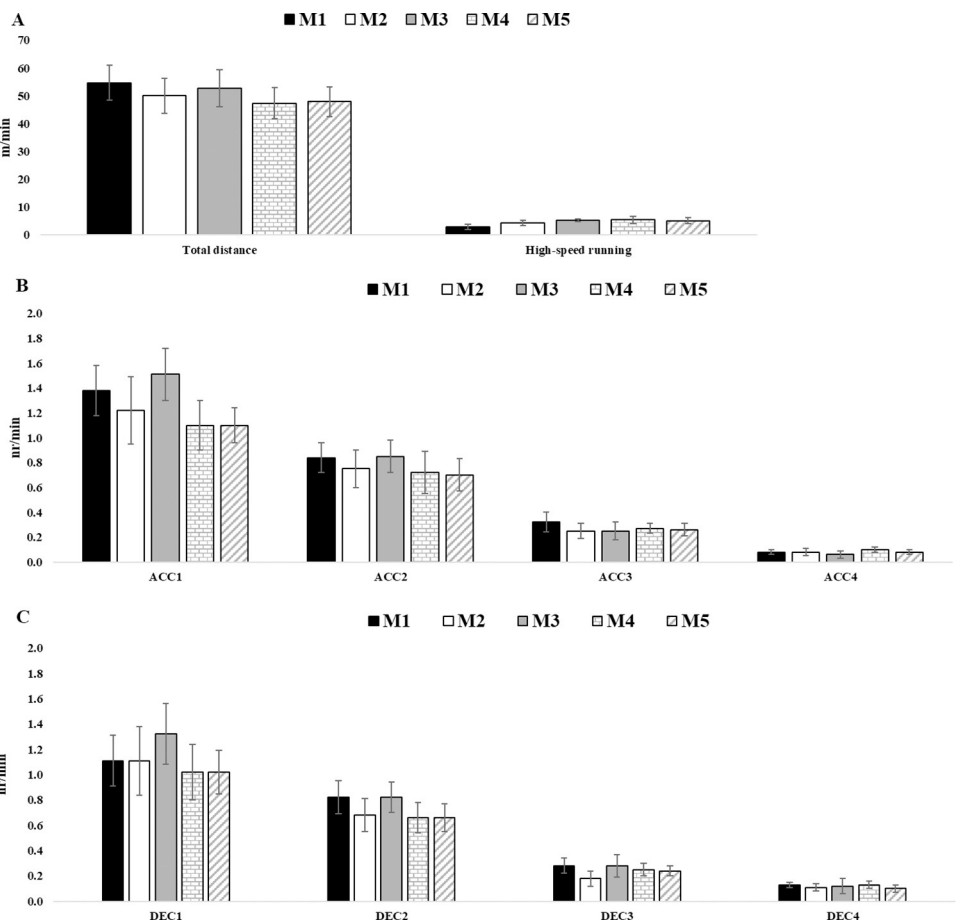

**Fig 1. Descriptive statistics for each GPS measure across the different microcycles (M1, M2, M3, M4, M5).** ACC1, acceleration 1; ACC2, acceleration 2; ACC3, acceleration 3; ACC4, acceleration 4; DEC1, deceleration 1; DEC2, deceleration 2; DEC3, deceleration 3; DEC4, deceleration 4.

**Table 1. Comparisons between M1 and M2.**

| Measures | M1 | M2 | % change | t-value | p-value | Effect size |
|---|---|---|---|---|---|---|
| TD (m/min) | 54.73±6.20 | 50.09±6.28 | -9.3 | 2.80 | **0.01** | 6.39 |
| HSR (m/min) | 2.94±0.88 | 4.33±1.01 | 32.1 | -5.77 | **<0.001** | 0.93 |
| ACC1 (nr/min) | 1.38±1.20 | 1.22±0.27 | -13.1 | 2.96 | **0.01** | 0.20 |
| ACC2 (nr/min) | 0.84±0.12 | 0.75±0.15 | -12.0 | 2.24 | **0.04** | 0.16 |
| ACC3 (nr/min) | 0.32±0.08 | 0.25±0.06 | -28.0 | 2.92 | **0.01** | 0.08 |
| ACC4 (nr/min) | 0.08±0.02 | 0.08±0.03 | 0.0 | 1.17 | 0.26 | 0.03 |
| DEC1 (nr/min) | 1.11±0.20 | 1.11±0.27 | 0.0 | -0.03 | 0.98 | 0.26 |
| DEC2 (nr/min) | 0.82±0.13 | 0.68±0.13 | -20.6 | 4.12 | **0.001** | 0.14 |
| DEC3 (nr/min) | 0.28±0.06 | 0.18±0.06 | -55.6 | 4.12 | **0.001** | 0.09 |
| DEC4 (nr/min) | 0.13±0.02 | 0.11±0.03 | -18.2 | 2.64 | **0.02** | 0.03 |

TD, total distance; HSR, high-speed running; ACC1, acceleration 1; ACC2, acceleration 2; ACC3, acceleration 3; ACC4, acceleration 4; DEC1, deceleration 1; DEC2, deceleration 2; DEC3, deceleration 3; DEC4, deceleration 4; Bold means significant result.

**Table 2. Comparisons between M2 and M3.**

| Measures | M2 | M3 | % change | t-value | *p*-value | Effect size |
|---|---|---|---|---|---|---|
| TD (m/min) | 50.09±6.28 | 52.68±6.70 | 4.9 | -1.04 | 0.32 | 9.65 |
| HSR (m/min) | 4.33±1.01 | 5.22±1.51 | 17.0 | -2.25 | **0.04** | 1.51 |
| ACC1 (nr/min) | 1.22±0.27 | 1.51±0.21 | 19.2 | -3.19 | **0.01** | 0.34 |
| ACC2 (nr/min) | 0.75±0.15 | 0.85±0.13 | 11.8 | -2.12 | 0.05 | 0.19 |
| ACC3 (nr/min) | 0.25±0.06 | 0.25±0.07 | 0.0 | 0.14 | 0.89 | 0.11 |
| ACC4 (nr/min) | 0.08±0.03 | 0.06±0.03 | -33.3 | 1.99 | 0.07 | 0.03 |
| DEC1 (nr/min) | 1.11±0.27 | 1.32±0.24 | 15.9 | -2.31 | **0.04** | 0.35 |
| DEC2 (nr/min) | 0.68±0.13 | 0.82±0.12 | 17.1 | -3.72 | **0.003** | 0.14 |
| DEC3 (nr/min) | 0.18±0.06 | 0.28±0.09 | 35.7 | -4.04 | **0.001** | 0.09 |
| DEC4 (nr/min) | 0.11±0.03 | 0.12±0.06 | 8.3 | -0.88 | 0.39 | 0.05 |

TD, total distance; HSR, high-speed running; ACC1, acceleration 1; ACC2, acceleration 2; ACC3, acceleration 3; ACC4, acceleration 4; DEC1, deceleration 1; DEC2, deceleration 2; DEC3, deceleration 3; DEC4, deceleration 4; Bold means significant result.

**Table 3. Comparisons between M3 and M4.**

| Measures | M3 | M4 | % change | t-value | *p*-value | Effect size |
|---|---|---|---|---|---|---|
| TD (m/min) | 52.68±6.70 | 47.39±5.57 | -11.2 | 2.53 | **0.03** | 8.05 |
| HSR (m/min) | 5.22±1.51 | 5.44±1.29 | 4.0 | -0.56 | 0.59 | 1.55 |
| ACC1 (nr/min) | 1.51±0.21 | 1.10±0.20 | -37.3 | 5.97 | <**0.001** | 0.26 |
| ACC2 (nr/min) | 0.85±0.13 | 0.72±0.17 | -18.1 | 3.14 | **0.01** | 0.16 |
| ACC3 (nr/min) | 0.25±0.07 | 0.27±0.04 | 7.4 | -0.85 | 0.41 | 0.08 |
| ACC4 (nr/min) | 0.06±0.03 | 0.10±0.02 | 40.0 | -4.85 | <**0.001** | 0.03 |
| DEC1 (nr/min) | 1.32±0.24 | 1.02±0.22 | -29.4 | 5.01 | <**0.001** | 0.23 |
| DEC2 (nr/min) | 0.82±0.12 | 0.66±0.12 | -24.2 | 5.04 | <**0.001** | 0.12 |
| DEC3 (nr/min) | 0.28±0.09 | 0.25±0.05 | -12.0 | 1.32 | 0.21 | 0.10 |
| DEC4 (nr/min) | 0.12±0.06 | 0.13±0.03 | 7.7 | -0.38 | 0.71 | 0.05 |

TD, total distance; HSR, high-speed running; ACC1, acceleration 1; ACC2, acceleration 2; ACC3, acceleration 3; ACC4, acceleration 4; DEC1, deceleration 1; DEC2, deceleration 2; DEC3, deceleration 3; DEC4, deceleration 4; Bold means significant result.

**Table 4. Comparisons between M4 and M5.**

| Measures | M4 | M5 | % change | t-value | *p*-value | Effect size |
|---|---|---|---|---|---|---|
| TD (m/min) | 47.39±5.57 | 47.89±5.40 | 1.0 | -0.25 | 0.81 | 7.76 |
| HSR (m/min) | 5.44±1.29 | 5.07±1.07 | -7.3 | 1.19 | 0.26 | 1.19 |
| ACC1 (nr/min) | 1.10±0.20 | 1.10±0.14 | 0.0 | -0.06 | 0.96 | 0.17 |
| ACC2 (nr/min) | 0.72±0.17 | 0.70±0.13 | -2.9 | 0.52 | 0.61 | 0.13 |
| ACC3 (nr/min) | 0.27±0.04 | 0.26±0.05 | -3.8 | 0.66 | 0.52 | 0.04 |
| ACC4 (nr/min) | 0.10±0.02 | 0.08±0.02 | -25.0 | 2.90 | **0.01** | 0.02 |
| DEC1 (nr/min) | 1.02±0.22 | 1.02±0.17 | 0.0 | 0.12 | 0.90 | 0.19 |
| DEC2 (nr/min) | 0.66±0.12 | 0.66±0.11 | 0.0 | -0.80 | 0.93 | 0.10 |
| DEC3 (nr/min) | 0.25±0.05 | 0.24±0.04 | -4.2 | 0.46 | 0.65 | 0.05 |
| DEC4 (nr/min) | 0.13±0.03 | 0.10±0.03 | -30.0 | 3.22 | **0.01** | 0.03 |

TD, total distance; HSR, high-speed running; ACC1, acceleration 1; ACC2, acceleration 2; ACC3, acceleration 3; ACC4, acceleration 4; DEC1, deceleration 1; DEC2, deceleration 2; DEC3, deceleration 3; DEC4, deceleration 4; Bold means significant result.

loads, fundamental for the development of explosive strength were included. When designed appropriately where space, duration and game rules are manipulated, these types of drills have previously proven to have a large physical component [26,27] that are ideal soccer-specific activities during the pre-season and in-season. A recent study analyzed female soccer teams training Ms and reported that the highest incidence of accelerations/decelerations was produced on MD-5 training days for non-starters and MD-4 for starters [28]. Therefore, further analysis into the training differences of daily external load is warranted as recently suggested [4].

Regarding HSR (>15km/h) load during pre-season, the highest values were achieved during M4, with slightly lower values during M5 (in-season). In female soccer competitions, players can cover 40 to 64% of the total distance in HSR (>15km/h), highlighting the importance of players being adequately prepared to cope with these match demands by increasing staff knowledge and tailoring training to replicate these demands [29]. The present results provide information suggesting there is a gradual increase in HSR distances across pre-season, aimed at establishing close to competitive match-play in order to positively impact HSR performance during match-play. Previous research has found improvements in performance following a tailored week of gradual reduction in HSR/sprint training load, thus reducing the volume of 30-m sprint distance [25].

The second study objective was to analyze the external load variation from M to M. When M1 and M2 were compared, statistically significant differences in almost all external load variables, except ACC4 and DEC1, were observed. However, the positive variation was only reported in HSR, while in all other variables, there was a reduction in external load from M1 to M2. Furthermore, it should be noted that only total distance presented a very large effect size. These results can potentially be explained by M1 consisting of 10 training sessions and one match, while during M2, five training sessions and one match were completed. When comparing M2 and M3, a positive and significant increase in HSR, ACC1, DEC1, DEC2 and DEC3 were reported. Similarly, from M3 to M4, an increase in neuromuscular activity, namely in the external load variable ACC4 was found, while a decreases in ACC1, ACC2, DEC1 and DEC2 occurred. The total distance covered was also significantly reduced from M3 to M4, again as a potential tapering strategy [24,25].

During the transition from pre-season (M1—M4) to in-season (M5), there was a general reduction in training load, most significantly in high-intensity accelerations and decelerations. During the pre-season Ms (M1—M4), the training emphasis was on increasing the players' physical condition [30,31], complemented with individualized gym-based strength training (referred to in the methodology). The training intensity was achieved mainly through small-sided games, and was gradually increased from M1 until the last pre-season week (M4) in order to expose players to these match-realistic actions to optimize players' match performance at the beginning of the official competition [24,25]. The last week of pre-season (M4) was significantly modified to reduce all external load metrics in order to super-compensate players in preparation for the initial competition period (M5) [14].

Despite the findings of the current study, there are some limitations that should be highlighted. The number of Ms from the in-season period (one) is less than the number of pre-season Ms (four). Only one senior female soccer team with 14 players was analyzed, suggesting that the results are only associated with the examined team, thus all findings should be cautiously interpreted, and the generalization to different leagues/countries must be carefully considered [15]. However, the study findings can be an important contribution for coaches, since the results of the present study reflect methodological decisions of planning and periodization in a senior female soccer team, considering players' load during the pre-season phase in preparation for optimal performance during in-season competition.

Future studies should include a greater number of Ms at different phases of the season and should include the evaluation of contextual variables known to affect physical output [32] and other soccer specific data such as technical and tactical [33]. Previous research in female soccer showed that accumulated training data, namely, distance at HSR, average speed, number of accelerations and decelerations was meaningfully higher when the next match was played at home, when compared with an away match. The same study also reported that the same external load measures were significantly higher when the next result was a loss, which contrasted with data from wins or draws [32]. However, a recent study in English Premier League male soccer players showed that more high-intensity efforts were performed in a 3-5-2 formation when compared with a 4-3-3 formation [33]. Such findings should also be considered in future designs examining elite female soccer players. Finally, the design of the present study can be extended to analyze different female teams across seasonal periods, as well as varying age groups of both female and male soccer.

## Conclusions

This study showed that in the first three pre-season Ms, players covered more total distance, number of ACC1, ACC2, DEC1 and DEC2 than in M4 and M5. Moreover, there was an increasing load from M1 to M4 for HSR and from M1 to M3 for ACC1, ACC2, DEC1 and DEC2. Furthermore, from M3 to M4, total distance, HSR, ACC1, ACC2, DEC1, DEC2, DEC3 decreased. While ACC4 and DEC4 decreased from M4 to M5, the other measures remained similar.

Therefore, an increase in training intensity as pre-season progressed was found, thus stabilizing the initial competitive M (M5) and providing a smooth transition into the initial competitive M, that had the objective of achieving optimal performance in the first official competition were found.

## Supporting information

**S1 File.**
(XLSX)

## Author Contributions

**Conceptualization:** Rafael Oliveira.

**Data curation:** Rafael Oliveira, Renato Fernandes.

**Formal analysis:** Rafael Oliveira.

**Investigation:** Rafael Oliveira, Ryland Morgans, Renato Fernandes, João Paulo Brito, Mário C. Espada, Fernando J. Santos.

**Methodology:** Rafael Oliveira, Renato Fernandes.

**Project administration:** Rafael Oliveira.

**Software:** Rafael Oliveira.

**Supervision:** Rafael Oliveira, João Paulo Brito, Mário C. Espada, Fernando J. Santos.

**Validation:** João Paulo Brito, Mário C. Espada, Fernando J. Santos.

**Writing – original draft:** Rafael Oliveira, Ryland Morgans, Renato Fernandes, João Paulo Brito, Mário C. Espada, Fernando J. Santos.

**Writing – review & editing:** Rafael Oliveira, Ryland Morgans, Renato Fernandes, João Paulo Brito, Mário C. Espada, Fernando J. Santos.

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
