## [Decision Letter · Decision Letter 0]

22 Oct 2024

PONE-D-24-31830External Load Transition Practices from Pre-Season to In-Season. A Case Study in Elite Female Professional Soccer PlayersPLOS ONE

Dear Dr. Oliveira,

Thank you for submitting your manuscript to PLOS ONE. After careful consideration, we feel that it has merit but does not fully meet PLOS ONE’s publication criteria as it currently stands. Therefore, we invite you to submit a revised version of the manuscript that addresses the points raised during the review process.

We look forward to receiving your revised manuscript.

Kind regards,

Rabiu Muazu Musa, PhD

Academic Editor

PLOS ONE

Journal Requirements:

Additional Editor Comments (if provided):

**Comments to the Author**

1. Is the manuscript technically sound, and do the data support the conclusions?

Reviewer #1: Yes

Reviewer #2: Yes

2. Has the statistical analysis been performed appropriately and rigorously? 

Reviewer #1: Yes

Reviewer #2: Yes

3. Have the authors made all data underlying the findings in their manuscript fully available?

Reviewer #1: Yes

Reviewer #2: Yes

4. Is the manuscript presented in an intelligible fashion and written in standard English?

Reviewer #1: Yes

Reviewer #2: Yes

5. Review Comments to the Author

Reviewer #1: The manuscript with the title “External Load Transition Practices from Pre-Season to In-Season. A Case Study in Elite Female Professional Soccer Players” presents results of an original research. It is quite relevant to state of art despite being a case study of one soccer team in which 14 players were included.

I do not know how sufficient is this number of athletes included in this study, authors may present this information as one of limitation of the paper.

Furthermore, this is a second part of the previous study (Oliveira et a., 2023) [doi:10.3390/medicina59122156]. I confirm and i see that both studies are different, that the current manuscript is relevant, novel and its results were not published elsewhere. To my knowledge, this approach was not applied before (both female and male players) which supports my positive opinion about it. The work can be replicated in other teams and sports, and it provides relevant information about external load variations from week to week, about the first 5 microcycles of the season (M1-M4 from pre-season and M5 from in-season).

The introduction provides a proper background and rationale to support the present research. The aim of the study is clear but it can be revised in terms of writing/maybe a much clearer reformulation.

Materials and methods can be fully replicated, and standard ethics were fulfilled. Statistics and results were properly applied and clearly presented.

Discussion is well addressed but there maybe a room for improving limitations and future directions. For instance, limitations can include the small sample size (number of players included) while the paragraph about future studies could specify the information about contextual variables (providing examples might help).

The conclusion is presented in an appropriate fashion and are supported by the data.

The paper is presented in an intelligible fashion and is written in standard English (only minor details can be improved).

Reviewer #2: Delete the first sentence of the abstract and the main text (intro). The conclusions needs major revision form the abstract and the main text. Carefully, read again the text for English language.

6. PLOS authors have the option to publish the peer review history of their article (what does this mean?). If published, this will include your full peer review and any attached files.

Reviewer #1: No

Reviewer #2: No

---

## [Author Response · Author response to Decision Letter 0]

29 Oct 2024

Please see attachment for better clarity. Nonetheless, we copy all comments and answers provided below: 

Authors: Data Availability Statement: Due to issues of participant consent related to the new data protection law from 25th may, 2018 (Portugal), data will not be 

shared publicly. Interested researchers may contact the corresponding author.

3. Please review your reference list to ensure that it is complete and correct. If you have cited papers that have been retracted, please include the rationale for doing so in the manuscript text, or remove these references and replace them with relevant current references. Any changes to the reference list should be mentioned in the rebuttal letter that accompanies your revised manuscript. If you need to cite a retracted article, indicate the article’s retracted status in the References list and also include a citation and full reference for the retraction notice

Authors: To the best of our knowledge, there are no retracted papers in the reference list. 

Reviewer #1: The manuscript with the title “External Load Transition Practices from Pre-Season to In-Season. A Case Study in Elite Female Professional Soccer Players” presents results of an original research. It is quite relevant to state of art despite being a case study of one soccer team in which 14 players were included.

I do not know how sufficient is this number of athletes included in this study, authors may present this information as one of limitation of the paper.

Authors: Dear reviewer, thank you very much for your feedback. We have followed your suggestion and completed the following sentence in the limitations paragraph: “Only one senior female soccer team with 14 players was analyzed, suggesting that the results are only associated with the examined team, thus all findings should be cautiously interpreted, and the generalization to different leagues/countries must be carefully considered [16].”

 All changes were highlighted in the manuscript with tracked changes. 

Furthermore, this is a second part of the previous study (Oliveira et a., 2023) [doi:10.3390/medicina59122156]. I confirm and i see that both studies are different, that the current manuscript is relevant, novel and its results were not published elsewhere. To my knowledge, this approach was not applied before (both female and male players) which supports my positive opinion about it. The work can be replicated in other teams and sports, and it provides relevant information about external load variations from week to week, about the first 5 microcycles of the season (M1-M4 from pre-season and M5 from in-season).

Authors: We really appreciate your opinion and completely agree that this work can be replicated in other teams and sports. Actually, we intend to do so. Thank you

The introduction provides a proper background and rationale to support the present research. The aim of the study is clear but it can be revised in terms of writing/maybe a much clearer reformulation.

Authors: Thank you for your valuable opinion. The work was revised to improve the writing accordingly.

Materials and methods can be fully replicated, and standard ethics were fulfilled. Statistics and results were properly applied and clearly presented.

Authors: Thank you for the recognition.

Discussion is well addressed but there maybe a room for improving limitations and future directions. For instance, limitations can include the small sample size (number of players included) while the paragraph about future studies could specify the information about contextual variables (providing examples might help).

Authors: As mentioned before, the small sample size was added to the limitations and suggestions for future studies with examples were now added as suggested. Thank you

The conclusion is presented in an appropriate fashion and are supported by the data.

Authors: Thank you for your consideration. Still, we provide a change to improve more clarity on the findings.

The paper is presented in an intelligible fashion and is written in standard English (only minor details can be improved).

Authors: The work was proofread to improve the clarity of writing.

Reviewer #2: Delete the first sentence of the abstract and the main text (intro). The conclusions needs major revision form the abstract and the main text. Carefully, read again the text for English language.

Authors: Dear reviewer, thank you so much for your positive opinion. We became very glad about it. The first sentence of the abstract was removed, and the first sentence of introduction was readjusted. The conclusion was revised in both abstract and main text. Lastly, the work was proofread to improve the clarity of writing. 

Thank you

---

## [Editor Report · Decision Letter 1]

5 Nov 2024

External Load Transition Practices from Pre-Season to In-Season. A Case Study in Elite Female Professional Soccer Players

PONE-D-24-31830R1

Dear Dr. Oliveira,

We’re pleased to inform you that your manuscript has been judged scientifically suitable for publication and will be formally accepted for publication once it meets all outstanding technical requirements.

Kind regards,

Rabiu Muazu Musa, PhD

Academic Editor

PLOS ONE
---

## [Editor Report · Acceptance letter]

11 Dec 2024

PONE-D-24-31830R1 

PLOS ONE

Dear Dr. Oliveira, 

I'm pleased to inform you that your manuscript has been deemed suitable for publication in PLOS ONE. Congratulations! Your manuscript is now being handed over to our production team.

Kind regards, 

on behalf of

Dr. Rabiu Muazu Musa 

Academic Editor

PLOS ONE